# Symplectic Geometry Aspects of the Parametrically-Dependent Kardar–Parisi–Zhang Equation of Spin Glasses Theory, Its Integrability and Related Thermodynamic Stability [note 1]

**DOI:** 10.3390/e25020308

**Published:** 2023-02-07

**Authors:** Anatolij K. Prykarpatski, Petro Y. Pukach, Myroslava I. Vovk

**Affiliations:** 1Department of Computer Science and Telecommunication, Cracow University of Technology, 31-155 Krakow, Poland; 2The Institute of Applied Mathematics and Fundamental Sciences, Lviv Polytechnical University, 79000 Lviv, Ukraine

**Keywords:** dynamical systems, symplectic analysis, dark type flow, thermodynamic stability, the Kardar–Parisi–Zhang-type equation, parametric evolution flow, conservation law, Poisson structure, symplectic structure, Lax–Noether condition, asymptotic solution, complete integrability, optimal control problem, 11.10.Ef, 11.15.Kc, 11.10.-z, 11.15.-q, 11.10.Wx, 05.30.-d

## Abstract

A thermodynamically unstable spin glass growth model described by means of the parametrically-dependent Kardar–Parisi–Zhang equation is analyzed within the symplectic geometry-based gradient–holonomic and optimal control motivated algorithms. The finitely-parametric functional extensions of the model are studied, and the existence of conservation laws and the related Hamiltonian structure is stated. A relationship of the Kardar–Parisi–Zhang equation to a so called dark type class of integrable dynamical systems, on functional manifolds with hidden symmetries, is stated.

## 1. Introduction

More than two hundred years ago the famous Gaussian distribution of random equally distributed real quantities was discovered. This represented one of the greatest societal contributions in mathematics, which allowed the development of a robust theory to explain and analyze much of the randomness inherent in the physical world. Physical and mathematical systems, described in terms of Gaussian statistics, are usually said to be in the *Gaussian universality class*. This class is very wide, but is not all encompassing. For example, classical extreme value statistics or Poisson statistics better capture the randomness and severity of events, ranging from natural disasters to emergency room visits. Over the past decades much effort has focused on evolution processes understanding [1,2,3], which are not well described within the classically developed statistical universality classes. Owing to the great impact of mathematical results in probability, PDEs, and representation theory, in mathematical and, to a great extent, in statistical physics, a need to dig deep in the study of systems that are not typically behaving statistical systems was recognized.

Amongst the newly revealed universally behaving stochastic systems, directed polymers in random media and spin glasses, having characteristic thermodynamic parameters that display non-Gaussian limiting distributions and are described by means of the well known [4] Kardar–Parisi–Zhang equation, emerged:(1)$$\partial v/\partial t=\partial^{2}v/\partial x^{2}-u{\left(\partial v/\partial x\right)}^{2}/2:=K\left[v,u\right]$$
with respect to the temperature $t\in R_{+}$ evolution parameter,. These represent a new universality class, the KPZ universality class, where x∈R, a function $v\in C^{2}\left(R\times R_{+};R\right)\subset M_{v}$ describes the distribution function of a related random variable, and $u\in C^{2}\left(R\times R_{+};R\right)\subset M_{u}$ is a parameter, which can often be directly (heuristically) computed for a particular growth model from microscopic dynamics, specified by the statistical physics model under regard. It is also worth noting that, since it is known that the universality of the KPZ equation arises only [5] in the special case of the model of corner growth, it is essentially equivalent [6] to the free energy of the directed random continuous polymer model, which parametrically depends on the macroscopically determined functional variable $u\in M_{u}.$ As the related Gaussian distributing function, described by the classical diffusion evolution flow
(2)$$\partial v/\partial t=\;\partial^{2}v/\partial x^{2},$$
on a functional manifold $M_{v}\subset C\left(R;R\right)$ possessing *a priori* the only conservation law 1=∫v(x,t)dx for all $t\in R_{+}$, normalizing the total probability for any normalized initial data $v\left(\cdot ,0\right)\in M_{v},$ the KPZ Equation (Equation 1), in general, possesses no conservation law; if no condition is imposed on the functional parameter $u\in M_{u}$ and its long-time temperature dependence.

As is well known, a simple non-linear Langevin-type Equation (Equation 1) allows [7,8,9,10] the macroscopic thermodynamic properties of a wide variety of growth processes to be described, such as the Eden model, growth of an interface in a random medium, randomly-stirred fluids, dissipative transport in the driven–diffusion equation, the directed polymer problem in a random potential and the behavior of flux lines in superconductors. Due to its ubiquity, any advance in understanding [11] solutions to the KPZ equation is likely to have wide significance [1,2,3] in both the fields of nonequilibrium thermodynamics and dynamical theory disordered systems. As was demonstrated in [4,12,13] the KPZ equation describes the long-time thermodynamics of the spin glass substrate growth well. Owing to competition between the surface tension smoothing forces and internal aggregation phase state, there is a tendency to prefer the direction of the local normal to the surface, represented by the corresponding nonlinear term.

What is worth mentioning here is the fact that the parametrically-generalized spin glass growth KPZ Equation (Equation 1) makes it possible to look at its thermodynamics from the optimal control problem approach, as was recently observed in a very inspiring work [9]. In this latter work, the relationship of the Hamilton-Jacobi theory to describing spin glass growth distribution function with the KPZ Equation (Equation 1) was shown. In particular, the interesting fact that the initial condition for the limiting spin glass free energy distribution function is determined by the second order Hamilton-Jacobi equation, coinciding with the the KPZ equation, was mentioned. The latter suggests an interesting statistical physics analysis of the thermodynamic properties of the macroscopically determined parameter $u\in M_{u},$ on which strongly depends the corresponding solutions [11] to the KPZ Equation (Equation 1). This creates an opportunity to control the spin glass growth process and its thermodynamic stability.

Thus, the problem arises as to how to describe the corresponding evolution constraints
(3)$$\partial u/\partial t\overset{?}{=}F\left[v,u\right]$$
on a functional parameter $u\in M_{u}\subset C\left(R;R\right),$ so as ensure to the existence of a nontrivial hierarchy of conservation laws for the combined evolution system
(4)$$\left.\begin{array}{c} \partial v/\partial t=\partial^{2}v/\partial x^{2}-u{\left(\partial v/\partial x\right)}^{2}/2\\ \partial u/\partial t=F[v,u] \end{array}\right\}\;:=Q\left[v,u\right],$$
considered a smooth vector vield $Q:M_{v}\times M_{u}\to T\left(M_{u}\times M_{v}\right)$ on the combined functional manifold $M_{v}\times M_{u},$ and which could be used for normalizing the corresponding distribution function $v\in C^{2}\left(R\times R_{+};R\right)$ for suitably chosen initial data $v\left(\cdot ,0\right)\in M_{v}.$

To solve this problem effectively, we made use of the gradient–holonomic algorithm, motivated by symplectic geometry techniques on functional manifolds [14,15], recently devised for studying the integrability properties of nonlinear dynamical systems with hidden symmetries and, in part, related with the optimal control theory [16,17] approach to parametrically dependent processes. Being applied to the parametrically-extended Kardar–Parisi–Zhang Equation (Equation 1) this algorithm allowed one to state that it belongs to a so *called dark type class* [18,19,20,21] of integrable Hamiltonian dynamical systems on functional manifolds with hidden symmetry. Namely, the parametrically-extended Kardar–Parisi–Zhang system of Equation (Equation 4) reduces to the evolution flow
(5)$$\left.\begin{array}{c} v_{t}=v_{xx}-\;uv_{x}^{2}/2\\ u_{t}\;=-u_{xx}\;-\;{\left(u^{2}v_{x}\right)}_{x}/2 \end{array}\right\}$$
on the combined functional manifold $M_{v}\times M_{u}\;$ possesses an infinite hierarchy of the conserved quantities, providing thermodynamically stable spin glasses growth processes. Moreover, it is a Lax type integrable Hamiltonian generalization of the dark type evolution flow (Equation 1) with respect to a suitably constructed Poisson structure on the manifold $M_{v}\times M_{u}.$

This result, in particular, indicates that the parametrically-extended Kardar–Parisi–Zhang Equation (Equation 1) possesses a rich internal hidden symmetry, allowing it to immerse into an infinite hierarchy of Lax type, completely integrable, dynamical systems on a functional manifold. Moreover, it also demonstrates that the Kardar–Parisi–Zhang Equation (Equation 1) allows an infinite hierarchy of completely integrable many-parametric dark type extensions, which can present some interesting applications to describing thermodynamical properties of polymers, structures in random media and spin glasses. From this point of view it is also interesting that the thermodynamically-based spin glass growth process control can analyze a dependence of a thermodynamically quasistable substrate diffusion parameter $r\in C^{2}\left(R\times R_{+};R\right)\subset M_{r}$ subject to the internal aggregation state, giving rise to the following physically more correct non-linear Langevin type equation:(6)$$\partial v/\partial t=\partial \left(r\partial v/\partial x\right)\partial x-u{\left(\partial v/\partial x\right)}^{2}/2$$
on the smooth functional manifold $M_{v},$. the solutions of which, in general, are strongly governed by the following temperature evolutions:(7)$$u_{t}\overset{?}{=}P\left[v.u,r\right],\;r_{t}\overset{?}{=}R\left[v.u,r\right]\;$$
for some smooth vector fields $P:M_{u}\to T\left(M_{u}\right)$ and $R:M_{r}\to T\left(M_{r}\right).$ Based on our approach to analyzing the parametrically-extended Kardar–Parisi–Zhang Equation (Equation 1), one can effectively describe the evolutions (Equation 7), under which the generalized non-linear Langevin type Equation (Equation 6) possesses a rich hierarchy of nontrivial conservation densities. This is strongly useful for controlling the related polymer and spin glass growth processes and their thermodynamic stability, and is considered a topic of our near future research.

## 2. Integrability Testing Algorithm

Consider now a nonlinear dynamical system (Equation 4) on a smooth 2π-periodic functional manifold $M:=M_{v}\times M_{u},$ which we assume to possess an infinite hierarchy of invariant densities and is *a priori* Lax type [14,22] integrable. The latter, in particular, means, as follows from the analysis in [15], that this infinite hierarchy of conservation laws is suitably ordered by powers a complex parameter λ∈C and has a respectively complexified generating gradient vector function $\varphi :=\varphi \left[v,u;\lambda \right]\in T^{*}\left(M\right)⊗C$, satisfying the Noether–Lax differential–functional equation
(8)$$\partial \varphi /\partial t+Q^{\prime ,*}\varphi =0.$$
As follows from the expression (Equation 8), the gradient vector function $\varphi \;\in T^{*}\left(M\right)⊗C$ possesses [23,24] the following asymptotic as λ→∞ solution
(9)$$\varphi ={\left(1,a_{1}\left(x;\lambda \right),a_{2}\left(x;\lambda \right),\ldots ,a_{m-1}\left(x;\lambda \right)\right)}^{⊺}\exp\left[\lambda^{s(\sigma )}t+\partial^{-1}\sigma \left(x;\lambda \right)\right],$$
where
(10)$$\sigma \left(x;\lambda \right)∼\sum_{j\in Z_{+}}\sigma_{j}\left[u\right]\lambda^{-j+s(\sigma )},\;a_{n}\left(x;\lambda \right)∼\sum_{j\in Z_{+}}\sigma_{j}\left[v,u\right]\lambda^{-j+s_{n}}\;$$
for some integers $s\left(\sigma \right),s_{n}\in Z_{+},n=\bar{1,m-1},$ which can be easily obtained upon substitution of the asymptotic solution (Equation 9) into (Equation 8). The representation (Equation 9), in particular, generates *a priori* an infinite hierarchy of functionals
(11)$$\gamma_{j}=\int_{0}^{2\pi }\sigma_{j}\left[v,u\right]dx,$$
$j\in Z_{+},$ on the manifold M, being conservation laws for the nonlinear dynamical system (Equation 4). The latter can be effectively used to construct additional structures inherent in this dynamical system. In particular, if a functional $H\in D\left(M\right)\in {\mathrm{span}}_{R}$$\left\{\gamma_{j}\in D\left(M\right):j\in Z_{+}\right\}$ can be represented in the following scalar form: $H=(\psi |u_{x}),$ and the element $\psi \in T^{*}\left(M\right)$ satisfies the modified Noether–Lax differential–functional equation
(12)$$\partial \psi /\partial t+Q^{\prime ,*}\psi =\mathrm{grad}L$$
for some smooth functional L∈D(M)， where the condition $\psi^{\prime }\neq \psi^{\prime ,*}$ should hold. The nonlinear dynamical system (Equation 4) proves to be *a priori* Hamiltonian with respect to the symplectic structure $\vartheta^{-1}=\psi^{\prime }-\psi^{\prime ,*}:$$T\left(M\right)\to T^{*}\left(M\right)$ and is represented as the flow
(13)$$u_{t}=-\vartheta \mathrm{grad}\left[\left(\psi |Q\right)-L\right]$$
on the functional manifold M. This makes it possible, in many practically important cases, to obtain a next compatible Poisson structure η:$T^{*}\left(M\right)\to T\left(M\right)$ on the manifold M, making it possible to construct the related Lax type representation for the nonlinear dynamical system (Equation 4).

### An Optimal Control Problem Aspect

Thedre is considered a smooth nonlinear dynamical system
(14)$$v_{t}=K\left[v,u\right]$$
on a 2π-periodic functional manifold $M_{v}\subset C\left(R/\left\{2\pi Z\right\};R^{m(v)}\right)$, depending parametrically on a 2π-periodic functional variable $u\in M_{u}\subset C\left(R/\left\{2\pi Z\right\};R^{m(u)}\right).$ Then, one poses the following Bellman–Pontriagin-type optimal control problem [16,17] for some smooth Lagrangian density $L:J_{\left(v,u\right)}\left(R;R^{2}\right)\to R$ on a temporal interval $\left[0,T\right]\subset R_{+}:$(15)$$v=\underset{v\in M_{v}}{\arg\inf}\int_{0}^{T}dt\int_{0}^{2\pi }L\left[v,u\right]dx$$
for a fixed functional parameter u∈M under the constraint that the evolution flow (Equation 14) possesses a smooth conserved quantity $\gamma =\int_{0}^{2\pi }\gamma \left[u,v\right]dx\in D\left(M_{v}\times M_{u}\right),$ that is dγ/dt=0 on the combined manifold $M_{v}\times M_{u}$ for all t∈[0,T]. The latter, in particular, means that we need to determine such an additional evolution flow
(16)$$u_{t}\overset{?}{=}F\left[v,u\right]$$
on the combined control manifold $M_{v}\times M_{v},$ which ensures the existence of the mentioned conserved quantity $\gamma \in D(M_{v}\times M_{u}).$ To solve this problem there it is suggested to analyze the extended Lagrangian functional
(17)$$\begin{array}{ccc} L[v;\mu ,\psi ] & : & =\int_{0}^{T}dt\left[\int_{0}^{2\pi }L\left[v,u\right]dx+\left(\psi |v_{t}-K\left[v,u\right]\right)\right.+\\ & & +\left.\mu \left(t\right)(\mathrm{grad}\gamma \left[u,v\right]|{\left(K\left[v,u\right],F\left[u,v\right]\right)}^{⊺})\right],\; \end{array}$$
supplemented with Lagrangian multipliers $\mu \in C_{0}^{1}\left(\left[0,T\right];R\right)$ and $\psi \in C_{0}^{1}\left(\left[0,T\right];T^{*}\left(M_{v}\right)\right)$ almost everywhere with respect to the temporal parameter t∈[0,T], and, next, to study its critical points:(18)$$\begin{array}{c} \delta \;L\left[v;\mu ,\psi \right]=0∼{\mathrm{grad}}_{v}L\left[v,u\right]-\left(\psi_{t}+K_{v}^{{}^{\prime ,*}}\left[v,u\right]\psi \right)+\\ +\mu \left(t\right)(\partial \mathrm{grad}\gamma /\partial t+{\left(K^{\prime ,*}\left[v,u\right],F^{\prime ,*}\left[v,u\right]\right)}^{⊺}\mathrm{grad}\gamma )=0 \end{array}$$
for all $\left(v,u\right)\in M_{v}\times M_{u}$ jointly with the condition that dγ/dt=0 for  t∈[0,T]. The obtained functional relationship (Equation 18) reduces to the following generalized Noether–Lax condition
(19)$$\psi_{t}+K_{v}^{{}^{\prime ,*}}\left[v,u\right]\psi =\mathrm{grad}\;L\left[v,u\right]$$
on the Lagrangian multiplier $\;\psi \in \;C_{0}^{1}\left(\left[0,T\right];T^{*}\left(M_{v}\right)\right)\subset T^{*}\left(M_{v}\right),$ as the following Noether–Lax equality
(20)$$\partial \mathrm{grad}\gamma /\partial t+{\left(K^{\prime ,*}\left[v,u\right],F^{\prime ,*}\left[v,u\right]\right)}^{⊺}\mathrm{grad}\gamma =0$$
holds *a priori* for any smooth conservation law $\gamma \in D(M_{v}\times M_{u})$ of the joint dynamical system
(21)$$v_{t}=K\left[v,u\right],\;v_{t}=F\left[v,u\right]$$
on the combined manifold $M_{v}\times M_{u}.$

A solution $\psi \in T^{*}\left(M_{v}\right)$ to the condition (Equation 19) allows the unique representation as the direct sum $\psi =\bar{\psi }⊕\varphi$ of its skew symmetric $\bar{\psi }\in T^{*}\left(M_{v}\right)$ and strictly symmetric $\varphi \in T^{*}\left(M_{v}\right)$ components, satisfying, respectively, the following differential–functional equations:(22)$${\bar{\psi }}_{t}+K_{v}^{\prime ,*}\left[v,u\right]\bar{\psi }=\mathrm{grad}\;L\left[v,u\right],$$
where, by definition, ${\bar{\psi }}_{v}^{\prime }\neq {\bar{\psi }}_{v}^{\prime ,*}$ on $M_{v}\times M_{u},$ and
(23)$$\varphi_{t}+K_{v}^{{}^{\prime ,*}}\left[v,u\right]\varphi =0,$$
where, by definition, $\varphi_{v}^{\prime }=\varphi_{v}^{\prime ,*}$ on $M_{v}\times M_{u}.$ Under the *a priori* assumed condition that the evolution flow (Equation 14) is a Hamiltonian system on the functional manifold $M_{v}$ with respect to the related symplectic structure $\Omega \in T^{*}\left(M_{v}\right)\wedge T^{*}\left(M_{v}\right),$ the differential-functional Equation (Equation 22) is always [14,25,26] solvable, giving rise to the known differential–geometric relationship
(24)$$\Omega ={\bar{\psi }}_{v}^{\prime }-{\bar{\psi }}_{v}^{\prime ,*},$$
subject to which the following compatible vector field representation
(25)$$K=-\Omega^{-1}\mathrm{grad}\;\left[\left(\;\bar{\psi }|K\right)-L\right]$$
holds on $M_{v}\times M_{u}.$ Simultaneously, the differential–functional Equation (Equation 23) is also always [14,25,26] solvable under the condition that φ=grad$\;\gamma \in T^{*}\left(M_{v}\right)$ for some conserved quantity $\gamma \in D(M_{v}\times M_{u})$ of the evolution flow (Equation 14) regardless of whether the evolution flow (Equation 14) on $M_{v}$ is Hamiltonian or not.

The differential–functional Equation (Equation 23) can be considered, in general, as a linear evolution partial differential equation on the complexified vector-function $\varphi \in \;T^{*}\left(M_{v}\right)⊗C,$ which always possesses [23,24] an asymptotic as λ→∞ solution (Equation 9) with *a priori local smooth densities* $\sigma_{j}\left[v,u\right]\in C^{\infty }(J(R/\left\{2\pi Z\right\};$$R^{m(v)}\times R^{m(u)});R),j\in Z_{+},$ which are conserved, that is $\frac{d}{dt}\int_{0}^{2\pi }\sigma_{j}\left[u,v\right]dx=0$ for all $j\in Z_{+}.$ Additionally, the evolution flow (Equation 14) on the functional manifold $M_{v}$ is *a priori* assumed to possess the conserved quantity $\gamma \in D(M_{v}\times M_{u}),$ its gradient $\varphi :={\mathrm{grad}}_{v}\gamma \left[u,v\right]$$\in T^{*}\left(M_{v}\right)$ necessarily satisfies the Noether–Lax differential-functional Equation (Equation 23), ensuring the existence of its asymptotic solution in the form (Equation 9). The latter condition, as the first step, makes it possible to regularly check the existence of the hidden symmetries mentioned above. In particular, local conserved quantities, by a second step, construct the searched additional evolution flow (Equation 16), solving our optimal control problem (Equation 15). Moreover, if the conserved quantities obtained this way prove to be suitably ordered, this case strictly corresponds to the completely integrable evolution flow
(26)$$\left.\begin{array}{c} v_{t}=K\left[v,u\right]\\ u_{t}=F\left[v,u\right] \end{array}\right.:=Q\left[v,u\right]$$
on the extended functional manifold $M_{v}\times M_{u}$. Below we apply the described symplectic geometry-based and optimal control problem motivated integrability problem solving algorithm to a new interesting class of the so called dark type nonlinear dynamical systems, the analytical studies of which were initiated by B. Kupershmidt [20,21], who demonstrated their interesting hidden symmetry and other related mathematical properties.

## 3. Hidden Symmetry Analysis of the Parametrically-Dependent Nonlinear
Kardar–Parisi–Zhang Equation

### 3.1. The Noether–Lax Equation and Its Asymptotic Solutions

To analyze the existence of the conserved quantities for the parametrically-dependent KPZ- evolution system (Equation 4)
(27)$$\left.\begin{array}{c} \partial v/\partial t=\partial^{2}v/\partial x^{2}-u{\left(\partial v/\partial x\right)}^{2}/2,\\ \partial u/\partial t=F[v,u] \end{array}\right\}\;,$$
on the 2π-periodic manifold $M_{v}\times M_{u},$ we will make use of the symplectic geometry based algorithmic scheme, devised in [14,15,27,28,29], within which, in particular, one needs to study special asymptotic λ→∞ solutions to the following Noether–Lax equation
(28)$$\varphi_{t}+K_{v}^{{}^{\prime ,*}}\left[v,u\right]\varphi =0$$
on the vector $\varphi \in T^{*}\left(M_{v}\right)⊗C,$ where, by definition, $\;K_{v}^{{}^{\prime ,*}}\left[v,u\right]:T^{*}\left(M_{v}\right)\to T^{*}\left(M_{v}\right)$ denotes the adjoining mapping [14,25,26] subject to the natural bilinear form $\left(\cdot |\cdot \right):T^{*}\left(M_{v}\right)\times T\left(M_{v}\right)\to R$ to the Frechet derivative mapping $\;\;K_{v}^{{}^{\prime }}\left[v,u\right]:T^{*}\left(M_{v}\right)\to T^{*}\left(M_{v}\right)$ with respect to the variable $v\in M_{v}.$ In our case the Equation (Equation 1) looks as follows:(29)$$\varphi_{t}+\varphi_{xx}+{\left(uv_{x}\right)}_{x}\varphi +uv_{x}\varphi_{x}=0,\;$$
the asymptotic λ→∞ solution [23,24,30,31] solution is representable as
(30)$$\varphi =\exp\left(-\lambda^{2}t+\partial^{-1}\sigma \right),\sigma ∼\sum_{j\in Z_{+}\cup \left\{-1\right\}}\sigma_{j}\left[v,u\right]\lambda^{-j}.$$ Having substituted the solution (Equation 30) into (Equation 29), one can easily obtain an infinite hierarchy of the following recurrent relationships:(31)$$-\delta_{j,-2}+\partial^{-1}\sigma_{j,t}+\sigma_{j,x}+{\left(uv_{x}\right)}_{x}\delta_{j,0}+uv_{x}\sigma_{j}+\sum_{k\in Z_{+}\cup \left\{-2,-1\right\}}\sigma_{j-k}\sigma_{k}=0,$$
where $\delta_{j,k},$$j,k\in Z_{+}\cup \left\{-2,-1\right\}$ denote the Koronecker symbols and $\partial^{-1}\left(\ldots \right):=$$\frac{1}{2}\left[\int_{0}^{x}\left(\ldots \right)ds-\int_{x}^{2\pi }\left(\ldots \right)ds\right]$ is the inverse to the differentiation ∂/∂x operator: $\partial /\partial x\cdot \partial^{-1}=1$ for all x∈R/{2πZ}. The solution of the relationships (Equation 31) gives rise to the following conserved quantities:(32)$$\sigma_{-1}=1,\sigma_{0}=-uv_{x}/2,\;\sigma_{1}=\frac{1}{4}\left[\partial^{-1}{\left(uv_{x}\right)}_{t}-{\left(uv_{x}\right)}_{x}+{\left(uv_{x}\right)}^{2}/2\right],\ldots$$
which can be successively continued, if one takes into account that the density $\sigma_{0}=-uv_{x}/2$ should be *a priori* conserved. The latter makes it possible to easily state that all the densities $\left\{\sigma_{j}:j\in N\right\}$ are conserved, if, to put by definition, that
(33)$${\left(uv_{x}\right)}_{t}={\left(uv_{x}\right)}_{xx}-uv_{x}\;{\left(uv_{x}\right)}_{x}.$$ This means, in particular, that the resulting system of two evolution Equations (Equation 1) and (Equation 33) possesses no other local conservation laws and is a Lax type integrable of the dark type. Now, taking into account the evolution Equation (Equation 4), we obtain the evolution flow we were looking for on the functional variable $u\in M_{u}:$(34)$$u_{t}=u_{xx}-1/2uu_{x}v_{x}+2u_{x}v_{xx}v_{x}^{-1}.$$ Thus, we can formulate the following proposition.

**Proposition** **1.**
*The parametrically-extended Kardar–Parisi–Zhang equation*

(35)
$$\left.\begin{array}{c} v_{t}=v_{xx}-uv_{x}^{2}/2\\ u_{t}=u_{xx}-1/2uu_{x}v_{x}+2u_{x}v_{xx}v_{x}^{-1} \end{array}\right\}:=Q\left[v,u\right]$$

*on the combined functional manifold $M_{v}\times M_{u}$ possesses only one local conserved quantity*

(36)
$$\gamma_{0}=\int uv_{x}dx,$$

*providing the thermodynamically stable spin glass growth process, and presenting a Lax type linearized generalization of the dark type evolution flow (Equation 1).*


**Proof.** The Lax type integrability of the combined Kardar–Parisi–Zhang Equation (Equation 35) easily follows from the extended Noether–Lax equation
(37)$$\psi_{t}+Q^{\prime ,*}\psi =0,$$
for an element $\psi \in T^{*}\left(M_{v}\times M_{u}\right)⊗C,$ having asymptotic λ→∞ as a solution
(38)$$\begin{array}{cc} \psi & ={\left(1,a\right)}^{⊺}\exp\left(-\lambda^{2}t+\partial^{-1}\sigma \right),\\ \sigma & ∼\sum_{j\in Z_{+}\cup \left\{-1\right\}}\sigma_{j}\left[v,u\right]\lambda^{-j},a∼\sum_{j\in Z_{+}}a_{j}\left[v,u\right]\lambda^{-j}, \end{array}$$
generates the same as the above truncated hierarchy of conserved densities $\{\sigma_{j}=0:j\in N\}.$ The latter gives rise, following the scheme devised in [32,33], to the Lax type linearization of the nonlinear dynamical system (Equation 35). □

We can observe here that the functional parameter $u\in M_{u}$ satisfies a perturbed diffusion type Burgers evolution equation, which represents some hidden physical properties of the related KPZ Equation (Equation 1).

### 3.2. Conserved Quantities and Dark Type Parametric Extensions of the
Kardar–Parisi–Zhang Equation

Consider now the following from (Equation 32) first conserved quantity:(39)$$\sigma_{1}=\frac{1}{4}\left[\partial^{-1}{\left(uv_{x}\right)}_{t}-{\left(uv_{x}\right)}_{x}+{\left(uv_{x}\right)}^{2}/2\right],$$
which makes it possible to put, by definition, that
(40)$${\left(uv_{x}\right)}_{t}=p_{x}-3/2uv_{x}{\left(uv_{x}\right)}_{x},$$
where an additional parametric variable $p\in M_{p}\subset C\left(R;R\right).$ The latter, in particular, means that if $M_{p}\subset \left\{p:J_{\left(u,v\right)}\left(R;R^{2}\right)\to R\right\},$ then the related evolution flow
(41)$$u_{t}=F\left[v,u\right]$$
should satisfy the following from (Equation 40) the differential–functional condition:(42)$${\left(Fv_{x}\right)}^{\prime ,*}\left(1\right)\;+{\left(uv_{xxx}-uu_{x}v_{x}^{2}/2-u^{2}v_{x}v_{xx}\right)}^{\prime ,*}\left(1\right)=0,$$
easily reducing to the next system of differential–functional relationships:(43)$$\begin{array}{ccc} F_{v}^{\prime ,*}v_{x}-F_{x}-u_{xxx}-{\left(uu_{x}v_{x}\right)}_{x}\; & = & 0,\\ F_{u}^{\prime ,*}v_{x}+v_{xxx}-uv_{x}v_{xx} & = & 0 \end{array}$$
on the manifold $M_{v}\times M_{u}.$ The second differential–functional relationship of (Equation 43) is easily solved as
(44)$$\begin{array}{ccc} F_{u}^{{}^{\prime ,*}} & = & -\partial^{2}/\partial x^{2}+\alpha uv_{x}\partial /\partial x+\left(1-\alpha \right)/2u\partial /\partial x\circ v_{x}\Rightarrow \\ & \Rightarrow & F_{u}^{\prime }=-\partial^{2}/\partial x^{2}-\alpha \partial /\partial x\circ uv_{x}+\left(\alpha -1\right)/2v_{x}\partial /\partial x\circ u\Rightarrow \\ & \Rightarrow & F\left[v,u\right]=-u_{xx}-\alpha {\left(u^{2}v_{x}/2\right)}_{x}+\left(\alpha -1\right)/2v_{x}uu_{x}, \end{array}$$
where α∈R is some parameter. Substitution of the result (Equation 44) into the first differential–functional relationship of (Equation 43) gives rise to the constraint α=1, thus ensuring the existence of the smooth evolution flow
(45)$$u_{t}\;=-u_{xx}\;-\;{\left(u^{2}v_{x}\right)}_{x}/2:=F\left[v,u\right],$$
on the functional manifold $M_{u},$ *a priori* generating, owing to the recurrence relationships (Equation 30), an infinite hierarchy of the nontrivial conservation laws: (46)$$\begin{array}{cc} \gamma_{0} & =\int uv_{x}dx,\gamma_{1}:=\int \sigma_{1}\left[u,v\right]dx=-\frac{1}{8}\int \left(u^{2}v_{x}^{2}+4u_{x}v_{x}\right)dx,\\ \gamma_{2} & :=\int \sigma_{2}\left[u,v\right]dx=\frac{1}{2}\int x\sigma_{1,t}\left[u,v\right]dx,\;\ldots ,\gamma_{j+1}:=\frac{1}{2}\int x\sigma_{j,t}\left[u,v\right]dx,\ldots \end{array}$$
for all j∈N. Thus, the obtained result can be formulated as the next proposition.

**Proposition** **2.**
*The parametrically-extended Kardar–Parisi–Zhang system of equations*

(47)
$$\left.\begin{array}{c} v_{t}=v_{xx}-\;uv_{x}^{2}/2\\ u_{t}\;=-u_{xx}\;-\;{\left(u^{2}v_{x}\right)}_{x}/2 \end{array}\right\}:=Q\left[v,u\right]$$

*on the combined functional manifold $M_{v}\times M_{u}$ possesses an infinite hierarchy of the conserved quantities*

(48)
$$\begin{array}{cc} \gamma_{0} & =\int uv_{x}dx,\gamma_{1}:=\int \sigma_{1}\left[u,v\right]dx=-\frac{1}{8}\int \left(u^{2}v_{x}^{2}+4u_{x}v_{x}\right)dx,\\ \gamma_{2} & :=\int \sigma_{2}\left[u,v\right]dx=\frac{1}{2}\int x\sigma_{1,t}\left[u,v\right]dx,\;\ldots ,\gamma_{j+1}:=\frac{1}{2}\int x\sigma_{j,t}\left[u,v\right]dx,\ldots \end{array}$$

*for all j∈N, providing thermodynamically stable spin glass growth process, and presents a Lax type integrable Hamiltonian generalization of the dark type evolution flow (Equation 1):*

(49)
$${\left(v_{t},u_{t}\right)}^{⊺}=-\vartheta \mathrm{grad}H\left[v,u\right],$$

*where $H=\int (u_{x}v_{x}+\;u^{2}v_{x}^{2}/4)dx$ is its Hamiltonian function and*

(50)
$$\vartheta =\left(\begin{array}{cc} 0 & 1\\ -1 & 0 \end{array}\right)$$

*denotes the corresponding Poisson structure on the manifold $M_{v}\times M_{u}.$*


**Proof.** The Lax type integrability of the combined dynamical system (Equation 47) easily follows from the extended Noether–Lax equation
(51)$$\psi_{t}+Q^{\prime ,*}\psi =0,$$
for an element $\psi \in T^{*}\left(M_{v}\times M_{u}\right)⊗C,$ whose asymptotic as λ→∞ solution
$$\psi ={\left(1,a\right)}^{⊺}\exp\left(-\lambda^{2}t+\partial^{-1}\sigma \right),\sigma ∼\sum_{j\in Z_{+}\cup \left\{-1\right\}}\sigma_{j}\left[v,u\right]\lambda^{-j},a∼\sum_{j\in Z_{+}}a_{j}\left[v,u\right]\lambda^{-j},$$
generates an infinite hierarchy of the nontrivial conserved densities $\{\sigma_{j}:J_{\left(u,v\right)}\left(R;R^{2}\right)\to R:j\in Z_{+}\},$ coinciding exactly with those (Equation 46), derived above. Following the scheme from [32,33], one easily derives the Lax type linearization of the nonlinear dynamical system (Equation 47).To state that the dynamical system (Equation 47) is Hamiltonian, we make use of the constructive algorithm, devised in [14,22,28] and based on the differential-geometric and symplectic structures of the conservation laws (Equation 48). Namely, let us consider the first nontrivial conservation law $\gamma_{0}\in D\left(M\right)$ and represent it in the following canonical Lagrangian form:
(52)$$\gamma_{0}=\int uv_{x}=\frac{1}{2}\int \left(uv_{x}-u_{x}v\right)dx=\left({\left(u,-v\right)}^{⊺}/2|{\left(v_{x},u_{x}\right)}^{⊺}\right):=-\left(\psi |{\left(v_{x},u_{x}\right)}^{⊺}\right),$$
where the vector $\psi =1/2{\left(-u,v\right)}^{⊺}\in T^{*}\left(M_{v}\times M_{u}\right)$ satisfies the following Noether–Volerra condition:
(53)$$L_{Q}\psi =\psi_{t}+Q^{\prime ,*}\psi ={\left(-u^{2}v_{x}\right)}_{x}/2,uv_{x}^{2}{/2)}^{⊺}=\mathrm{grad}L\left[v,u\right],$$
where $L_{Q}:T^{*}\left(M_{v}\times M_{u}\right)\to T^{*}\left(M_{v}\times M_{u}\right)$ denotes the usual Lie derivative with respect to the vector field $Q:M_{v}\times M_{u}\to T\left(M_{v}\times M_{u}\right),$ defined by the nonlinear dynamical system (Equation 47), and we put, by definition, the functional $\;L:=\int \left(u^{2}v_{x}^{2}/2\right)dx.$ Taking now into account that the Noether–Volerra condition (Equation 53) can be equivalently rewritten as
(54)$${\left(v_{t},u_{t}\right)}^{⊺}=Q\left[v,u\right]=-\vartheta \mathrm{grad}\left[\left(\psi |Q\right)-L\right]\left[v,u\right],$$
meaning the Hamiltonian representation of the parametrically-generalized Kardar–Parisi–Zhang evolution flow (Equation 47):
(55)$${\left(v_{t},u_{t}\right)}^{⊺}=-\vartheta \mathrm{grad}H\left[v,u\right],$$
where we put, by definition,
(56)$$\begin{array}{ccc} \vartheta^{-1} & : & =\psi^{\prime }-\psi^{\prime ,*}=\left(\begin{array}{cc} 0 & 1\\ -1 & 0 \end{array}\right),\\ H & : & =\left(\psi |Q\right)-L=\int \left(u_{x}v_{x}+\;u^{2}v_{x}^{2}/4\right)dx, \end{array}$$
having denoted by $\vartheta :T^{*}\left(M_{v}\times M_{u}\right)\to T\left(M_{v}\times M_{u}\right)$ the corresponding skew–symmetric Poisson operator on the combined functional manifold $M_{v}\times M_{u}$ and the related Hamiltonian function $H=-4\gamma_{1}\in D\left(M_{v}\times M_{u}\right)).$ The same way, having used the derived above hierarchy of conservation laws, one can construct other compatible with (Equation 56) Poisson structures on the functional manifold $\;M_{v}\times M_{u}$, ensuring the related Lax type linearization [14,22] of the parametrically-extended Kardar–Parisi–Zhang system of Equations (Equation 47). □

The corresponding scrutinized analysis of the infinite hierarchy of recurrent relationships (Equation 31) shows that the scheme developed above by expanding the KPZ evolution Equation (Equation 1) on additional parametric functional manifolds $M_{\overset{´}{p}}:=\;M_{p_{1}}\times M_{p_{2}}\times \ldots \times M_{p_{n}}$ can be successively continued by means of introducing new functional variables $p_{j}\in M_{p_{j}}\subset C\left(R;R\right),j=\bar{1,n},$ for arbitrary natural n∈N and for suitably determining the corresponding evolutions $p_{j,t}=P_{j}\left[v,u,p_{1},p_{2},\ldots ,p_{j}\right]$ on functional manifolds $M_{p_{j}},$ under which the joint dynamical system
(57)$$\begin{array}{ccc} v_{t} & = & v_{xx}-uv_{x}^{2}/2,u_{t}=F\left[v,u,p_{1}\right],\\ p_{1,t} & = & P_{1}\left[v,u,p_{1}\right],\ldots ,p_{n,t}=P_{n}\left[v,u,p_{1},p_{2},\ldots ,p_{n}\right] \end{array}$$
possesses an infinite hierarchy of suitably ordered conservation laws on the functional manifolds $M_{\overset{´}{p}}:=\;M_{p_{1}}\times M_{p_{2}}\times \ldots \times M_{p_{n}}$ and represents a nonlinear integrable dynamical system.

For instance, taking now into account that the density $\sigma_{0}:J_{\left(u,v\right)}\left(R;R^{2}\right)\to R$ is *a priori* a conserved quantity, its temporal derivative $\sigma_{0,t}:J_{\left(u,v,p\right)}\left(R;R^{3}\right)\to R$ can be represented as full ∂/∂x-derivative:(58)$$\;\;\;\;\sigma_{0,t}=-\frac{1}{2}\left(p_{x}-uv_{x}{\left(uv_{x}\right)}_{x}\right),\;\;{\left(uv_{x}\right)}_{t}={\left(A\left[v,u\right]p\right)}_{x}-uv_{x}{\left(uv_{x}\right)}_{x},$$
for some new functional variable $\;p\in M_{p}\subset C\left(R;R\right),$ whose unknown *A*
(59)$$p_{t}=P\left[v,u,p\right]$$
depends on the condition that the related density $\sigma_{1}:J_{\left(v,u,p\right)}\left(R;R^{3}\right)\to R$ is conserved, that is
(60)$${\left(A\left[v,u\right]p\right)}_{t}=\eta {\left[v,u,p\right]}_{x}\;$$
for some densities $A:J_{\left(v,u\right)}\left(R;R^{2}\right)\to R$ and $\eta :J_{\left(v,u,p\right)}\left(R;R^{3}\right)\to R.\;$ The latter is easily reduced to the following differential condition
(61)$$\begin{array}{ccc} {\left(A_{t}\right)}_{v}^{\prime ,*}p+A_{v}^{\prime ,*}P+P_{v}^{\prime ,*}A & = & 0,\\ {\left(A_{t}\right)}_{u}^{\prime ,*}p+A_{u}^{\prime ,*}P+P_{u}^{\prime ,*}A & = & 0,\\ {\left(A_{t}\right)}_{p}^{\prime ,*}\;p+A_{t}+A_{u}^{\prime ,*}P+P_{u}^{\prime ,*}A & = & 0 \end{array}$$
should hold under the following temporal evolutions
(62)$$\begin{array}{ccc} v_{t} & = & v_{xx}-uv_{x}^{2}/2,p_{t}=P\left[v,u,p\right],\\ u_{t} & = & {\left(A\left[v,u\right]p\right)}_{x}v_{x}^{-1}\;-uv_{x}^{-1}v_{3x}-uu_{x}v_{x}/2 \end{array}$$
for all $\left(v,u,p\right)\in M_{v}\times M_{u}\times M_{p}.$ The obtained determining expressions (Equation 61) can be effectively dissolved subject to the evolution flow (Equation 59) under some suitably chosen density $A:J_{\left(v,u\right)}\left(R;R^{2}\right)\to R,$ as these expressions are looking too complicated for their analytical solution. A unique clue concerning the general form of the density $A:J_{\left(v,u\right)}\left(R;R^{2}\right)\to R$ consists in its dimensional symmetry, easily following from the third equation of the evolution Equation (Equation 62). For instance, if dim(A)= dim$(u_{x}^{2}v),$ then $A\left[v,u\right]=u_{x}^{2}v+c_{1}u_{x}^{2}+c_{2}u^{2}v_{x}^{2}+c_{3}u_{xx}/v+c_{4}u_{x}v_{x}uv^{-1}+\ldots ,$ where the coefficients $c_{j}\in R,j\in N,$ can be successively determined from the system (Equation 61) simultaneously with the corresponding expression for the evolution (Equation 59) by means of simple, yet slightly cumbersome, analytical calculations, which we plan to present within a separate work under preparation.

## 4. Conclusions

We sketched a symplectic geometric scheme of studying dark type nonlinear dynamical systems with hidden symmetries and applied it to analyzing the well known parametrically-dependent Kardar–Parisi–Zhang equation, describing spin glass growth dynamics. Its finitely-parametric evolution extensions, possessing a finite number of conserved quantities, and important for effective modeling of new quasi-stable materials, were constructed in detail by means of the differential–geometrically based gradient–holonomic and optimal control motivated integrability problem solving algorithms. A relationship between the parametrically-generalized Kardar–Parisi–Zhang type Hamiltonian flow to a so *called dark type class* of integrable dynamical systems on functional manifolds with hidden symmetries was described. The relationship possesses infinite hierarchies of nontrivial conserved quantities and is Lax type linearizable on a suitably extended functional manifold.

## Data Availability

Not applicable.

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
