# Peer review of "Symplectic Geometry Aspects of the Parametrically-Dependent Kardar–Parisi–Zhang Equation of Spin Glasses Theory, Its Integrability and Related Thermodynamic Stability†"

_entropy, 2023, doi:10.3390/e25020308_

Round 1
Reviewer 1 Report
Please answer the following questions:
1. In this paper, the first paragraph of the Introduction is not supported by references, so please add them for improvement.
2.The authors makes the same Proposition 1 and Proposition 3, so what is the connection and difference between them? Can this situation be explained?
3.There are several errors in the references, for example, the year is wrong in [28], etc. Please polish the manuscript carefully.
Reviewer 2 Report
In this work, the author studied a thermodynamically unstable spin glass growth model described by means of the parametrically dependent Kardar-Parisi-Zhang equation. It is analyzed within the symplectic geometry based gradient-holonomic and optimal control motivated algorithms. The finitely-parametric functional extensions of the model is studied, the existence of conservation laws and the related Hamiltonian structure is stated. The results are very interesting and warrant for publication, after minor revisions. It should be indicated clearly the proof of Proposition 1, namely, what is the beginning and the ending of such a proof.
Reviewer 3 Report
This paper uses symplectic geometry to analyze a thermodynamically unstable spin glass growth model. The latter is defined by a parametrically dependent Kardar-Parisi-Zhang (KPZ) equation.
Algorithms with gradient-holonomic and optimal control bases are implemented. The existence of conservation laws and Hamiltonian structure are demonstrated, and the KPZ equation is revealed to belong to a so-called dark type class of integrable dynamical systems on functional manifolds. The proof of Proposition 2, for example, supports this demonstration.
This paper presents some new analysis of a fundamental equation of mathematical physics, with an underlying basis in geometry and thermodynamics. It will be of interest to physicists and applied mathematicians working in these subject areas. Publication seems appropriate, but a few minor changes are recommended:
1) It would be interesting to include some more details and qualitative examples of dark-type systems (e.g., hidden symmetry), in the introduction, to appeal to a wider readership, especially applications to thermodynamical systems.
2) This reader did not follow the notation used in the recursive equations such as eq. (36). Please define all symbols carefully here.
3) On p. 8, a reference with more details on the Burgers type of evolution equations should be mentioned, though the general form of such an equation will be known to many mathematical physicists. But there are different particular forms depending on the applied field of study.
4) Page 7, the sentence below is very awkward and does not make sense, please revise it:
"Below we will apply the described above symplectic geometry based and optimal control problem motivated integrability problem solving algorithm to ..."
5) Top of p. 8, appears to be a missing "}" notation for the densities before eq. (38).
Round 2
Reviewer 1 Report
The comments have been carefully addressed by the authors, therefore I recommend the revised manuscript to be accepted as is.